# Nanoimprinted and Anodized Templates for Large-Scale and Low-Cost Nanopatterning

**DOI:** 10.3390/nano11123430

**Published:** 2021-12-17

**Authors:** David Navas, David G. Trabada, Manuel Vázquez

**Affiliations:** Instituto de Ciencia de Materiales de Madrid, ICMM-CSIC, 28049 Madrid, Spain; davidgonztra@gmail.com

**Keywords:** large-scale nanopatterning, combined imprint-electrochemical processes, porous anodic alumina membranes, square and triangular 2D templates, ferromagnetic antidots

## Abstract

Nanopatterning to fabricate advanced nanostructured materials is a widely employed technology in a broad spectrum of applications going from spintronics and nanoelectronics to nanophotonics. This work reports on an easy route for nanopatterning making use of ordered porous templates with geometries ranging from straight lines to square, triangular or rhombohedral lattices, to be employed for the designed growth of sputtered materials with engineered properties. The procedure is based on large-scale nanoimprinting using patterned low-cost commercial disks, as 1-D grating stamps, followed by a single electrochemical process that allows one to obtain 1-D ordered porous anodic templates. Multiple imprinting steps at different angles enable more complex 2-D patterned templates. Subsequently, sputtering facilitates the growth of ferromagnetic antidot thin films (e.g., from 20 to 100 nm Co thick layers) with designed symmetries. This technique constitutes a non-expensive method for massive mold production and pattern generation avoiding standard lithographical techniques. In addition, it overcomes current challenges of the two-stage electrochemical porous anodic alumina templates: (i) allowing the patterning of large areas with high ordering and/or complex antidot geometries, and (ii) being less-time consuming.

## 1. On the State-of-the-Art of Non-Lithographic Nanopatterning

During the last decades, the use of templates has been widely exploited for the fabrication of micro- and nanostructured materials. The development of lithographic techniques has nicely allowed the achievement of nanostructures with different shapes and sizes as small as a few tens of nm [1,2]. However, they require the use of sophisticated and expensive equipment, which makes it difficult to be incorporated into the high-throughput and mass production industry. Thus, the development of more simple and less expensive fabrication methodologies for industrial procedures is a current challenge.

Templates prepared by self-assembling have been proposed as a potential alternative to overcome these limitations [3]. Among the various self-assembling methodologies, anodization is an electrochemical process that has been widely used for the fabrication of oxidized nanostructured surfaces of different materials, such as Al_2_O_3_ [4,5,6], TiO_2_ [7,8], iron oxide [9,10] or SiO_2_ [11,12]. In particular, porous anodic alumina (PAA) membranes were shown to be suitable for large-scale and inexpensive production of 1-, 2- and even 3-dimensional well-controlled nanotemplates [4,5,6,13] and with technological applicability in a broad spectrum of research and industrial fields, such as in high-density magnetic storage, solar cells, gas sensors or drug delivery [14].

Since the mid-1990s [4], PAA have been fabricated using a well-established two-step anodization process [15,16,17,18,19]. While the first anodization step introduces the ordering degree by performing a pre-patterning of the aluminum (Al) substrate, the second is the one in which the template thickness can be controlled. On the other hand, the anodization parameters (applied voltage, temperature and the anodic solution) are engineered to determine the interpore distance as well as the pore diameter [16,17,18,19]. However, this self-assembled approach presents some disadvantages, such as the inability to obtain templates with neither a perfect degree of ordering at large scale nor complex geometries, and it is also a time-consuming process (up to several days).

Therefore, new and more efficient methodologies are demanded, such as hybrid nanofabrication techniques where lithographic and self-assembling methods are combined [20,21]. Among several approaches, an alternative to reach well-controlled templates consists of the mechanical nanoimprinting by which the pattern of a hard stamp (mold) is transferred onto a substrate by mechanical pressure. Nanoimprint lithography was first introduced in 1996 by Chou et al. [22], who created resist-based nanostructured templates by compressing a mold with a thin resist film, followed by an anisotropic etching to transfer the pattern through the entire template thickness.

Regarding the fabrication of PAA, this concept was successfully applied for the preparation of ordered nanoporous alumina templates after an anodization process of pre-patterned aluminum substrates that were previously imprinted by a SiC master mold prepared by electron beam lithography [23], or alternatively by Si_3_N_4_ [24,25] or nickel [26] master molds prepared by interference lithography. This methodology has been shown to be very effective for the preparation of defect-free alumina templates with different pore array ordering, from the standard hexagonal-array [23] to square- and triangle-array geometries [27,28]. Alternative techniques used for the pre-treatment of the Al substrate include ion-beam lithography [29], scanning probe microscopy [30], atomic force microscopy [31] and self-organized periodic array of polystyrene particles [32].

In addition, complex 2-D geometries, such as square, rectangular, rhombohedral or triangular patterns, have been successfully transferred onto different types of polymeric films using a simple 1-D grating stamp and subsequent multiple imprints [33]. There are few reports in the literature that have confirmed the possibility of applying this last approach for the fabrication of nanoporous alumina templates with different patterns employing a single lineal master mold, such as those based on a commercial optical grating [34] or a Ni line stamp [35], followed by several imprinting processes [34,35,36].

In this work, we introduce a non-lithographic, low-cost methodology to fabricate PAA-based large-scale templates using large nanoimprint molds followed by a single anodization process [37]. A novel aspect of this approach consists of using commercial compact discs as the imprint stamps to replicate their ordering onto the Al substrate instead of more conventional lithographic methods. Compact discs have been successfully used for different types of patterning and lithographic techniques. For example, CDs were used as imprint molds for the generation of nanostructured PMMA polymeric films [38,39] or grating-patterned TiO_2_ antireflection layers for perovskite solar cells [40]. The use of CDs has even supported the generation of color patterns on various surfaces [41] or the patterning of microstructures and nanostructures of soluble materials (known as lithographically controlled wetting) [42]. Here, we have demonstrated that the patterns of these molds can be successfully transferred onto an Al substrate by single or multiple imprint processes and can achieve complex 1- or 2-dimensional well-controlled alumina templates.

These templates can be suitably employed as precursors of long-range nanopatterned structures for applications in nano-electronics, spintronics or nano-photonic devices. Here, we have prepared and studied ferromagnetic antidot thin films, grown by sputtering on these PAA templates, and with technological applicability as magnonic crystals [43,44] and magneto-plasmonic devices [45,46]. Ferromagnetic antidot arrays are nanostructures with well-defined magnetization pinning centers that affect the magnetization reversal process by controlling the nucleation and propagation of domain walls [47,48]. Additionally, a rich variety of magnetization configurations in antidot arrays have been observed whose related magnetic properties can be easily tuned by tailoring the antidot lattice symmetry [49,50], hole diameter [51] and shape [52,53], inter-hole distance [54,55], film thickness [56] and lattice defects [57]. Although most of these works were carried out by using lithographic techniques, antidot thin films with both either in-plane [58,59,60] or out-of-plane magnetization easy axis [61,62,63,64,65] have also been reported using PAA templates. Therefore, we have grown and studied ferromagnetic antidot thin films with unconventional geometries such as linear or square structures.

In summary, we believe that our methodology, based on commercial products and conventional techniques, could be rapidly incorporated into the mass production industry with low cost and reduced time-consuming production.

## 2. Fabrication of Ordered Large Nanostructures by Combined Imprint and Anodization

Ordered nanostructures at large scale (up to a few cm^2^) have been fabricated by combined imprint and electrochemical anodization. Figure 1 represents a schematic view of the whole process involving the imprint with a commercial stamp on a pretreated Al substrate. Imprint can be either a single- or multi-step process if subsequent imprints are performed at different angles, producing several configurations, from linear to square, triangular or rhombohedral. Afterwards, the imprint patterned Al substrates are electrochemically anodized, giving rise to ordered porous structures. In the following, we describe in further detail the different preparation steps. At the bottom of Figure 1, the great versatility of the proposed methodology for the manufacture of nanostructured porous alumina patterns is shown, summarizing the different parameters that can be used, as well as examples showing linear, square, rhombohedral and triangular configurations.

### 2.1. Commercial Compact Discs as Nanoimprint Molds

Commercial compact discs have been used as imprint stamps to replicate their ordering onto the Al substrate as an alternative to more conventional lithographic methods. In 1980, companies such as Philips and Sony introduced the compact disc (CD) for storing digital data. Although it is a very useful medium for recording data, demand for new media with higher storage capacity led to the development of the digital versatile disc (DVD) during the 1990s, and later to the Blu-ray (BR) disks. All these media typically consist of a 120 mm diameter polycarbonate disk with a thickness of 1.2 mm. From a general point of view, they are made up of two main patterned sections of polycarbonate. For both DVD and CD media, one of the patterned polycarbonate sections is coated with a thin reflective metallic (Al) layer. Although the reflective coating is different for BR media, its physical structure is similar to both DVD and CD. In order to obtain our patterning stamps, and according to the protocol described in References [39,66], the disks were first cut into pieces of about (3 × 3) cm^2^. Later, commercial tape was glued to these pieces, and the polycarbonate sections were slowly peeled off and properly cleaned using methanol (see Figure 2).

There are several mass-produced brands offered in the market. Although available erasable/re-writable optical disks are not completely identical to each other, the clean stamps have similar linear gratings. In this work, we have used Verbatim CD and DVD [67], as well as Intenso BR [68]. Appendix A, collects the geometrical parameters, such as the linewidth, periodicity and height, of the used optical disks. To determine the morphological properties of the nanostructures, scanning electron microscopy (SEM) analysis has been performed using a Philips XL30-FEG-SEM microscope (FEI Company, Eindhoven, Netherlands). SEM images of the polycarbonate stamps with the metallic layer, obtained from DVD and BR-disks, are shown in Figure 2c,d, respectively.

### 2.2. Large-Scale Imprint-Patterned Al Foils

As starting material for single and multiple imprinting steps, we considered high-purity (99.999%) Al foils (diameter and thickness of 20 mm and 0.5 mm, respectively) degreased and electrochemically polished in a mixture of perchloric acid (HClO_4_) and ethanol (C_2_H_5_OH) to clean and reduce the surface roughness [17]. Afterwards, the features of the master mold were transferred to the Al substrate by an imprint process using a commercial hydraulic press and applying a pressure of 250 kg/cm^2^. Multiple imprinting steps at different angles enable the transfer of more complex 2-D patterned templates, such as square or rhombohedral patterns, when a second imprint step is performed with the mold rotated by an angle of 90° or 60°, respectively.

Figure 3 shows SEM images of the line arrays generated on the Al substrate after an imprint process using molds based on CD, DVD or BR disks, respectively. In agreement with the optical disks’ geometrical properties (described in Appendix A), the patterned Al substrates present periodicities of around 1600, 740 or 320 nm (see Figure 3a–c, respectively).

Square-based patterns can also be achieved when a second imprint step is performed with the mold rotated by an angle of 90° (examples are shown in Figure 4a–c). More complex patterns, such as rectangular (Figure 4d) or rhombohedral geometries (Figure 4e), can be produced by using different molds for each imprint step and when the mold is rotated for the second imprint process by 90° or 60°, respectively. Although we are using polymeric molds, we have made several imprints without the mold being degraded and/or destroyed by the applied pressure.

### 2.3. PAA Nanostructures Grown by Anodization of Imprint-Patterned Al Foils

After the single- or multi-imprint process, a single anodization step was conducted under a constant voltage of 195 V in 0.03 M phosphoric acid (H_3_PO_4_) solution at a temperature of 5° C for 2 h. We observed that the anodization process in such a diluted phosphoric acid-based solution, and for time ≤2 h, did not significantly alter the configuration of the pre-patterned aluminum foil.

As indicated above, three stamps, namely CD-PS/ML, DVD-PS/ML and BR-PS/ML molds, were employed. As an example, we focus on the results obtained using DVD-PS/ML mold while similar data were observed when CD-PS/ML or BR-PS/ML molds were used (see for example Appendix A). Figure 5 shows SEM images of the PAA nanostructured templates after a single anodization step of Al substrates pre-patterned using a DVD-PS/ML mold, with stripe, square and triangular geometries.

PAAs with a single-imprint process exhibit two main regions (Figure 5a–c): the alumina grown in the highest sections of the pre-patterned Al substrate show the formation of a continuous line due to the collapse of the pores (or frustrated pores), while the second region shows a well-aligned line of pores with an average pore diameter of ≈200 nm. A closer inspection at the cross-section SEM image (Figure 5c) shows that these well-aligned pores transform into two pores along alumina template thickness.

On the other hand, multi-imprint steps allow the achievement of more complex geometries. Two-imprint steps generate a well-ordered pore array with square configuration when the mold was rotated by an angle of 90° between imprints (Figure 5d–f). It was observed that PAA nanostructured templates with a single pore, with an average diameter of ≈70 nm, grew within each square of ≈500 nm lateral sizes. Finally, three-imprint steps, where the mold was rotated by an angle of 60° between imprints, were able to generate a triangular configuration with a more complex pore distribution (see Figure 5g–i).

In order to check the influence of the experimental conditions, the same pre-patterned Al substrates were also anodized under a lower constant voltage (165 V) in 0.03M H_3_PO_4_ solution at 5 °C and for 2 h. PAA templates with very similar morphologies were obtained (see Appendix A), confirming that the final configuration of the PAA templates is mainly controlled by the geometry of the pre-patterned Al substrates.

## 3. Sputtering Ordered Co Nanostructures on Imprint-Anodic PAA Templates

The described imprint-anodic samples can be used as templates for the generation of various nanostructures with encoded ordering for a wide spectrum of applications. As a particular example, we address the case of several Co antidot thin films and their overall magnetic characterization of interest for applications. Therefore, after the anodization process, the PAA nanostructured templates were coated with a Co layer using a homemade rf sputtering system. The Ar (99.999% pure) sputtering gas pressure was fixed to 1.2 × 10^−2^ mbar, the base pressure was below 2.5 × 10^−4^ mbar, and the power applied was 99 W for the 5 cm diameter target. Under these experimental conditions, the Co deposition rate was 3.1 nm/s. Three samples were prepared with 20, 50 and 100 nm thicknesses, respectively.

### 3.1. Morphology and Structural Characterization of Ordered Co Nanostripes and Square Arrays

The SEM images in Figure 6a,b correspond to the 50 nm thick sputtered Co sample. Longitudinal Co double-nanostripe structures are observed, showing a lateral periodical modulation replicating the previously imposed ordering of the anodic PAA nanostructure (shown in Figure 5a–c). The modulation is interpreted to be a consequence of the anodization process in connection with the observed transversal understructure (see Figure 6b) that corresponds to the Co growth on the groove zones connecting the Co double-nanostripe structures.

In order to grow ordered arrays of Co squares with cubic symmetry, we used the ordered nanostructured templates shown in Figure 5d–f. Then, the Co layer was sputtered onto them in a similar way as it was previously described. Figure 6c,d shows SEM images of the 50 nm thick Co squared dot array. Nanodots take near-squared shape (520 × 520 nm^2^), separated by around 220 nm with a pore at the center and a macroscopic squared arrangement. As it was observed in the case of Co nanostripes, we note that neighboring Co squared nanodots are connected by a Co understructure. This is confirmed in Appendix A where a freestanding nanoporous Co membrane was obtained after the chemical dissolution from the PAA template.

Microstructural characterization of ordered Co nanostructures was performed by X-ray diffraction (XRD) using a Bruker D8 Advance diffractometer (Bruker, Billerica, MA, USA) with Cu Kα radiation (λ = 1.540593 Å). The diffraction pattern (see Appendix A for additional details) shows three well-defined peaks corresponding to Al (220) and (311), as well as to Co at 2θ ≈ 43.74°. Since the fcc Co (1 1 1) reflection (2θ = 44.571°) [69] is close to the hcp Co (0 0 0 2) reflection (2θ = 44.949°) [70], the crystallographic phase of the Co layer cannot be resolved. Then, we suggest that the Co layer is polycrystalline and crystallized in hcp phase with (0 0 0 1) texture or fcc with (1 1 1) texture.

### 3.2. Magnetic Characterization of Ordered Co Nanostripes and Co Square Arrays

The magnetic characterization of these Co nanostructures was carried out at room temperature using a vibrating sample magnetometer (VSM, model KLA-Tencor EV7, KLA-Tencor Corporation, Westwood, MA, USA). The VSM experimental setup allowed the rotation of the sample so that it was possible to determine the azimuthal (in-plane) angular dependence under a maximum magnetic field of ±18 kOe.

As it was previously described [71], the in-plane magnetization reversal of a Co thin film mostly proceeds by the nucleation and propagation of domain walls, resulting in nearly square hysteresis loops and coercivity fields of ≈30 Oe. Figure 7 shows the azimuthal angular dependence of the hysteresis loops for the Co double-nanostripes with thickness of 20 (Figure 7a–c), 50 (Figure 7d–f) and 100 nm (Figure 7g–i). In all of them, the hysteresis loops, M/Ms vs. H, also show a nearly square shape with a single giant Barkhausen jump for a configuration of the applied field parallel to the nanostripes (0°). The loops evolve progressively to S-shaped with decreasing susceptibility as the orientation of the magnetic field rotates towards the perpendicular orientation (90°). The measurements indicate that maximum coercivity, H_c_, and reduced remanence, M_r_/M_s_, values are observed nearly along the direction parallel to the nanostripes, indicating that this corresponds to the magnetization easy axis in these three samples. The perpendicular orientation of the applied field roughly corresponds to a magnetization hard axis.

The angular dependence of coercivity for the 20 nm (Figure 7b) and 50 nm (Figure 7e) thick nanostripes can be interpreted assuming a strong, nearly uniaxial shape anisotropy originating in the high length-to-thickness aspect ratio of the nanostripes. However, an additional anisotropic term should be considered to understand the local angular coercivity maxima at around 60° and 120°. This effect is even more pronounced in the case of thicker nanostripes (100 nm thick) where the coercivity angular profile particularly reduces its anisotropic behavior (see Figure 7h). The occurrence of such multiple secondary maximum values for the coercivity indicates that the anisotropy is not fully uniaxial but contains a four-fold anisotropy. The origin for that four-fold symmetry seems to originate from the presence of the Co transversal understructures. Note however that for the reduced remanence angular profile (Figure 7c,f,i), such four-fold anisotropy is less apparent.

A preliminary magnetic characterization was also carried out in the Co squared nanodots arrays (see Appendix A). There, we observe a less remarkable angular dependence, which is related to a less defined shape anisotropy term. A deeper analysis of the angular dependence of coercivity and reduced remanence suggests the presence of a modest in-plane bi-axial magnetic anisotropy with magnetization easy axes at 90° to each other. The in-plane bi-axial magnetic anisotropy seems to be related to the cubic array of squared Co nanodots. Further analysis of complex geometrical nanostructures will be detailed in the future.

## 4. Conclusions

In summary, the main objective of this work has been the development of a methodology for the fabrication of large-scale and low-cost PAA nanostructured templates with well-defined order and versatile complex geometries. The procedure consists of a first step by which the ordering of commercially standard stamps such as CD, DVD and BR disks is transferred into Al foils. It is followed by an electrochemical single anodization process resulting in nanostructured porous ordered alumina templates. Moreover, it was demonstrated that complex 2-D template structures can be produced from 1-D stamps by multiple imprinting steps at different angles. This low time-consuming and inexpensive technique makes such nanostructured templates suitable for a gentle incorporation into the high-throughput and mass production industry.

These PAA nanostructures can be further used as templates to grow specific geometrical nanostructures such as arrays of nanostripes or squared/rombohedral ordered nanostructures. In particular, a specific study has been performed on ordered Co nanostripe thin films and squared dot arrays. It was confirmed that the magnetic behaviour of the sputtered antidot thin films mainly depends on the shape anisotropy contribution and can be engineered by the appropriate nanoimprint processes.

Therefore, the proposed technique constitutes a non-expensive method for massive mold production and pattern generation avoiding standard lithographical techniques [32]. A number of technologies, such as nano-photonic and nano-electronic devices, are expected to profit from this simple and cost-effective fabrication methodology.

## Figures and Tables

**Figure 1 nanomaterials-11-03430-f001:**
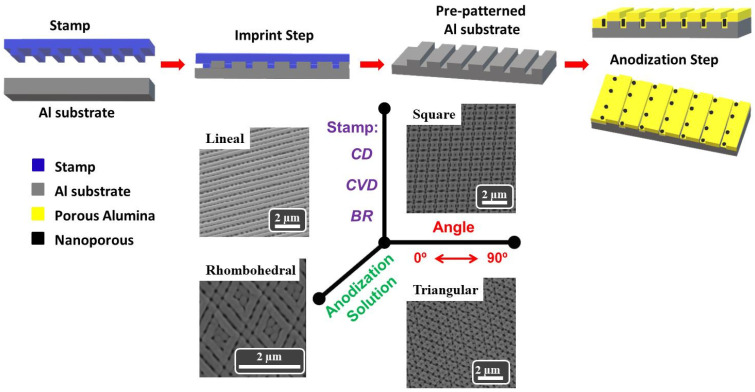
Schematic view summarizing the imprint and anodization processes to obtain the ordered porous nanostructured template. A general summary of the different parameters that can be used in the proposed fabrication process is shown at the bottom of the figure.

**Figure 2 nanomaterials-11-03430-f002:**
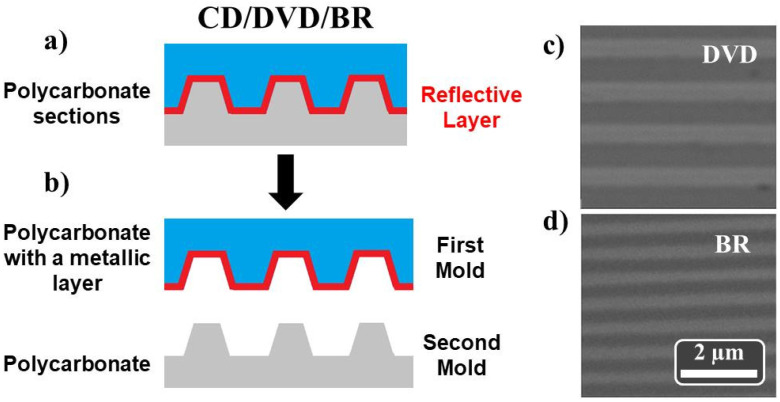
Schematic diagrams of the CD, DVD or BR disk media before (**a**) and after (**b**) the separation of the polycarbonate sections. SEM images of the polycarbonate stamp with a metallic layer obtained from (**c**) DVD and (**d**) BR disks.

**Figure 3 nanomaterials-11-03430-f003:**
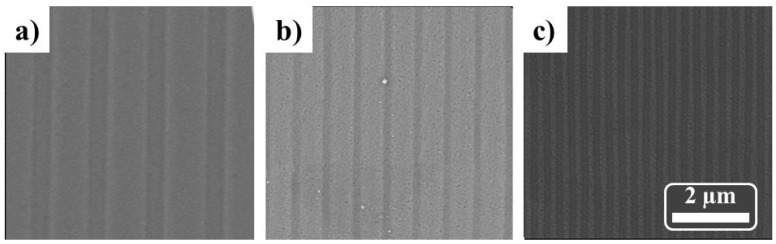
SEM images of stripe patterns on the Al substrate after one imprint step using CD-PS/ML (**a**), DVD-PS/ML (**b**) and BR-PS/ML (**c**) molds.

**Figure 4 nanomaterials-11-03430-f004:**
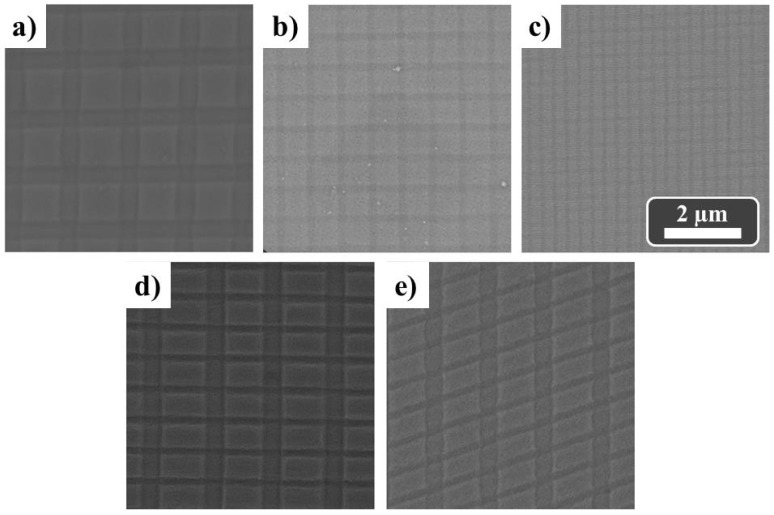
SEM images of square patterns on the Al substrate after two imprint steps using CD-PS/ML (**a**), DVD-PS/ML (**b**) and BR-PS/ML (**c**) molds. SEM images of rectangular and rhombohedral patterns on Al substrate after two imprint steps combining CD- PS/ML and DVD- PS/ML molds and when the mold was rotated for the second imprint process by 90° (**d**) and 60° (**e**).

**Figure 5 nanomaterials-11-03430-f005:**
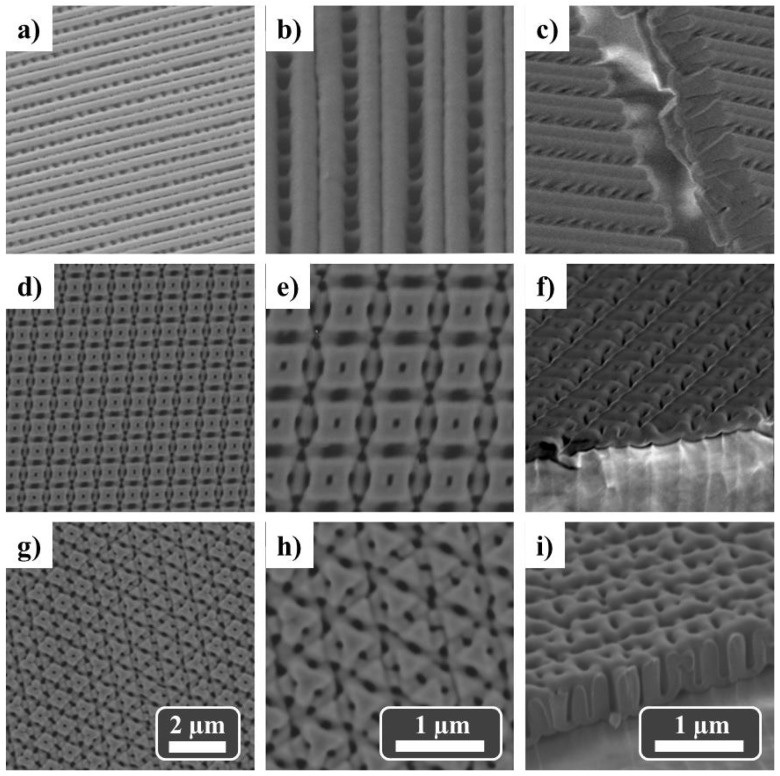
Top-view (**a**,**b**,**d**,**e**,**g**,**h**) and cross-section (**c**,**f**,**i**) SEM images of PAA templates with line (**a**–**c**), square (**d**–**f**) and triangular configurations (**g**–**i**) imprinted using DVD-PS/ML molds and a single anodization step under constant voltage of 195 V in 0.03M H_3_PO_4_ solution at 5 °C and for 2 h.

**Figure 6 nanomaterials-11-03430-f006:**
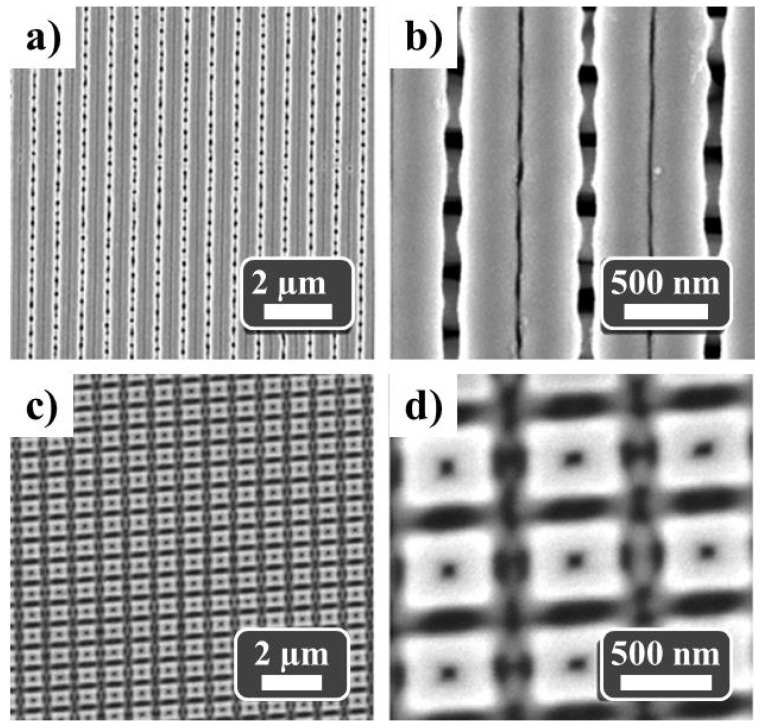
Top-view SEM images of (**a**,**b**) an array of Co double-nanostripes and (**c**,**d**) Co squares in a cubic arrangement (50 nm thick Co antidot films in both cases).

**Figure 7 nanomaterials-11-03430-f007:**
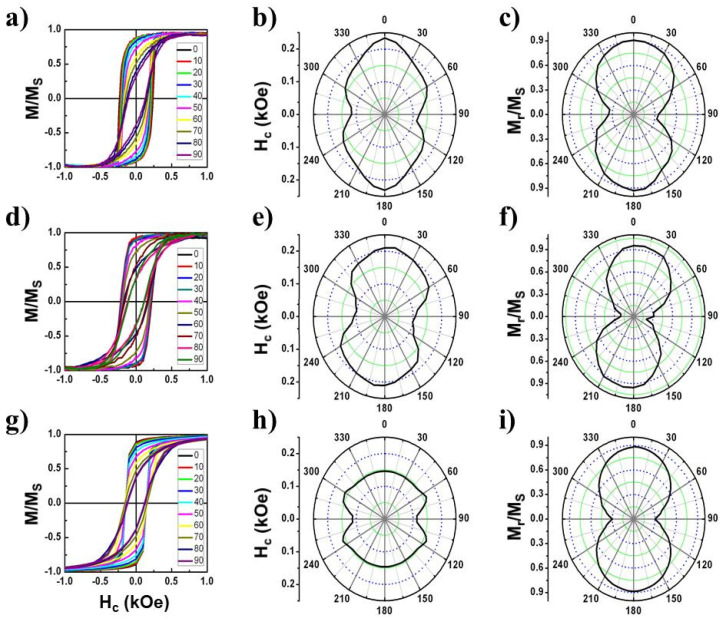
In-plane azimuthal VSM hysteresis loops, M/Ms, of ordered Co double-nanostripes with 20 (**a**), 50 (**d**) and 100 nm (**g**) Co thick. The corresponding angular dependence of coercivity, Hc, and reduced remanence, Mr/Ms, are given in (**b**,**e**,**h**) and (**c**,**f**,**i**), respectively. The reference 0° corresponds to parallel alignment of the applied field with the nanostripe main axis.

## Data Availability

The data presented in this study are available on request from the corresponding author.

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
