# Peer review of "Nanoimprinted and Anodized Templates for Large-Scale and Low-Cost Nanopatterning"

_nanomaterials, 2021, doi:10.3390/nano11123430_

Round 1

Reviewer 1 Report

The authors demonstrate nanopatterning of Al substrates using a two-step process, where the substrates are first imprinted by a mold consisting of a piece of an optical storage media (CD/CVD/BR), and subsequent anodization of the Al. Both square and hexagonal arrays could be made by subsequent imprints after rotation of the substrate. Finally, a ferromagnetic Co thin film was sputter coated onto the Al substrate, and the magnetization response was obtained using a VSM magnetometer.   

This is a very neat and interesting piece of experimental work that certainly warrants publication. The ms is well written, and the experimental results are nicely presented. One could perhaps have wished for a more clear SEM comparison of the substrate before and after the anodization step, and corresponding reference VSM measurements of non-anodized samples. Also, the amount of self-citations to own anodization work could be reduced without loosing important information.

Reviewer 2 Report

The manuscript by Navas et al. investigates a combination of nanoimprinting and anodic oxidation to fabricate an anodized large-scale nanopatterned surface using very low-cost tools.

In my opinion, the topic is exciting and fits the journal's scope. However, some issues need clarification and improvements before the manuscript to be acceptable for publication in Nanomaterials. Once addressed the points listed below, I expect the manuscript to be suitable for publication in Nanomaterials.

Remarks

As mentioned by the authors in the introduction, anodized nanostructured surfaces are very important in nanotechnology. The author limits their example to alumina; however, there are many other examples of different anodized materials used in nanoscience. For example, SiO2 nanostructures fabricated by local anodic oxidation are used as active elements in resistive switching (see Adv. Mater. 2012, 24, 1197-1201); this literature, or similar, should be mentioned in the introduction.

A similar comment consideration can be done about the use of low-cost tools by the authors for nanopatterning. The use of CD and DVD is undoubtedly a significant advantage; however, they are already used for several types of unconventional lithography (see, for example, Nat. Protocols 2012, 7, 1668-1676); this literature, or similar, should be mentioned in the introduction.

The title of section 2 is misleading. In my opinion, the authors do not describe a “synthesis” but a fabrication process.

The authors claim the use a polycarbonate stamp to imprint a metal.  This interpretation sounds very strange because of the difference in the hardness of the two materials.  This reviewer does not contest the experimental result, but he is perplexed by the interpretation.  Is it possible that the authors deposit a small polycarbonate layer on the surface during the imprinting step? 

It could work as a mask during the imprinting step. The authors should comment on this point.

An Atomic force microscopy image in-phase mode (i.e. sensible to the hardness of the sample) of the printed substrate could help the authors clarify this point.

I suggest also adding an image of the cross-section of the printed substrate before the anodization.

The application of such a high voltage is unusually for electrochemistry in water. Have the voltage a role in the organization of the forming nanopores?

Round 2

Reviewer 2 Report

The authors addressed all the remark of the previous evaluation, this reviewer has not further comment.